# Inkjet Printing of Functional Electronic Memory Cells: A Step Forward to Green Electronics

**DOI:** 10.3390/mi10060417

**Published:** 2019-06-22

**Authors:** Iulia Salaoru, Salah Maswoud, Shashi Paul

**Affiliations:** Emerging Technologies Research Centre, De Montfort University, Hawthorn Building, The Gateway, Leicester LE1 9BH, UK; P12049171@email.dmu.ac.uk (S.M.); spaul@dmu.ac.uk (S.P.)

**Keywords:** inkjet, printing, functional materials, silver, PEDOT:PSS, memory cells, green processes

## Abstract

Nowadays, the environmental issues surrounding the production of electronics, from the perspectives of both the materials used and the manufacturing process, are of major concern. The usage, storage, disposal protocol and volume of waste material continue to increase the environmental footprint of our increasingly “throw away society”. Almost ironically, society is increasingly involved in pollution prevention, resource consumption issues and post-consumer waste management. Clearly, a dichotomy between environmentally aware usage and consumerism exists. The current technology used to manufacture functional materials and electronic devices requires high temperatures for material deposition processes, which results in the generation of harmful chemicals and radiation. With such issues in mind, it is imperative to explore new electronic functional materials and new manufacturing pathways. Here, we explore the potential of additive layer manufacturing, inkjet printing technology which provides an innovative manufacturing pathway for functional materials (metal nanoparticles and polymers), and explore a fully printed two terminal electronic memory cell. In this work, inkjetable materials (silver (Ag) and poly(3,4-ethylenedioxythiophene)-poly(styrenesulfonate) (PEDOT:PSS)) were first printed by a piezoelectric Epson Stylus P50 inkjet printer as stand-alone layers, and secondly as part of a metal (Ag)/active layer (PEDOT:PSS)/metal (Ag) crossbar architecture. The quality of the individual multi-layers of the printed Ag and PEDOT:PSS was first evaluated via optical microscopy and scanning electron microscopy (SEM). Furthermore, an electrical characterisation of the printed memory elements was performed using an HP4140B picoammeter.

## 1. Introduction

Inkjet printing technology is an additive manufacturing technique that works by generating small ink droplets and propelling those droplets onto a substrate. This digital printing technology consists of three main features: ink, printhead and substrate. First, from the ink perspective the chemical and physical properties of the solution play a crucial role, mainly in the formation and ejection of the ink. Secondly, the appropriate printhead (thermal and piezoelectric) and a suitable driving voltage waveform (the width and amplitude of the pulse) are essential for accurate, reliable and reproducible drop ejection. Furthermore, equally important in the printing process is the substrate, which has a strong impact on achieving high quality printed patterns. There has been immense and continuous interest in the digital printing technique, which can be attributed to its cost effectiveness, efficacy of material use, compatibility with a wide range of substrates, digital and additive deposition, maskless nature and suitability for small to large area deposition. Some of these advantages are presented in Figure 1.

In a nutshell, digital printing technology, which is defined as “the printing of things (PoTs)”, is capable of revolutionizing the whole system of manufacturing functional materials and electronic devices, therefore creating a “digital industrial revolution” or a pathway to large area and high throughput flexible electronics. As part of this digital revolution, both inorganic and organic polymers have been exploited. Printing conductive metal nanomaterial inks, such as copper (Cu), gold (Au), silver (Ag) and aluminium (Al) inks, is one of the directions that has been extensively investigated and the core findings have been presented in comprehensive review papers [1,2,3]. Copper [4,5,6] and aluminium [7] are highly abundant metals, but unfortunately suffer from rapid oxidation at room temperature in air. These metals are still being used in the electronics industry as contact materials, and the deposition is carried out in an inert environment. Thus, the interest in the use of such metals for digital printing is diminishing. As we know, gold [8,9,10] is a very good conductor and does not have an oxidation issue, but is rather expensive. Due to the aforementioned problems, nanoparticle based silver ink [11,12,13,14] has attracted special attention, due to its stability, compatibility with the required ink additives, relatively high resistance to oxidation, high electrical conductivity and very competitive price. Hence, based on all of these factors we can safely conclude that silver is currently the best metal for printing conductive patterns. Such patterns can be used in flexible and large area electronics. Organic materials are also currently being studied as an alternative to metallic conductors for printing conductive paths, for example polypyrroles [15], graphene [16,17,18,19], polyanilines [19] and poly(3,4-ethylenedioxythiophene)-poly(styrenesulfonate) PEDOT:PSS [20,21].

As we have discussed previously, digital printing can bring a number of benefits to the electronics industry. The printing of electronic memory is a thriving field that attracts the interest of not only the academic community, but industry as well. From the academic perspective, Hubber et al. [22] reported fully inkjet printed stand-alone resistive memory cells. In another report, Porro and Ricciardi [23] fabricated an asymmetric metal-insulator-metal (MIM) type memristor based on graphene oxide, in which only the active layer (graphene oxide) was deposited via inkjet printing and the top Al contact was deposited using a conventional thermal vacuum evaporation method. On the other hand, from an industrial perspective, a Norwegian company, Thinfilm, developed a flexible, rewritable, ferroelectric, polymer-based memory and recently this technology was transferred to Xerox Corp for up-scaling and manufacturing.

In this study, all components (the electrodes and the active core of the two terminal devices) were fabricated using digital inkjet printing technology, and hence a fully printed crossbar memory array was achieved. To realise such a pathway, the main physical properties of the silver and PEDOT:PSS based inks, such as the viscosity, surface tension and pH, and their wettability on both rigid and flexible substrates were first investigated. Secondly, both Ag and PEDOT:PSS stand-alone multi-layers were printed using an Epson Stylus P50, a commonly used desktop printer (Epson, Suwa, Nagano, Japan) and the quality and surface thickness were evaluated by optical microscopy and scanning electron microscopy (SEM). Furthermore, adhesive tape tests were performed in order to understand the adhesion properties of the printed patterns on the selected substrates. Finally, a two terminal Ag/PEDOT:PSS/Ag crossbar structure was prototyped using inkjet printing alone, and in-depth electrical characterisations (i.e., current–voltage characteristic and memory retention time tests) were performed using a HP4140B-pico-ammeter (Keysight, Santa Rosa, CA, USA).

## 2. Experimental Section

The silver and PEDOT:PSS inks were evaluated and deposited via inkjet printing. The silver nanoparticle (AgNP) ink (Drycure Ag-j) was a mixture of 8–22% silver by weight, 18–52% water by weight, 20–65% glycerol by weight and a small amount of alcohol. The PEDOT:PSS ink was purchased from Sigma-Aldrich (St. Louis, MO, USA), and was composed of 2% PEDOT:PSS in an ethylene glycol monobutyl ether:water (3:2) mixture. In order to validate that the inks were suitable for printing, their main properties such as the surface tension, viscosity and pH, were tested. The surface tension of the inks was measured using a torsion balance model “OS” (Weston-super-Mare, UK) and the viscosity was evaluated by a Brookfield DV2T viscometer (Brookfield, Toronto, ON, Canada). In addition, a 3520 pH meter was used to test the pH of the inks.

Secondly, in order to ensure good adhesion of the ink to the substrate, the wetting behaviour of both inks on rigid and flexible substrates was evaluated by measuring the ink/substrate contact angle via the sessile drop analysis method (Attention Theta Optical Tensiometer, Biolin Scientific, Gothenburg, Sweden).

A commercial desktop inkjet printer, the Epson Stylus P50, was employed to deposit Ag and PEDOT:PSS multi-layer patterns. The Epson Stylus P50 is a piezoelectric printer with a printhead that has 90 nozzles with a 65 μm nozzle diameter, 1.5 pL drop size and 360 dpi resolution. Both the printer and silver ink were purchased from Printed Electronics Limited (PEL) (Tamworth, UK). Moreover, PEL optimised the ink to match the requirements of the printer.

*Printing functional materials based ink*. Both the Ag and PEDOT:PSS inks were kept in an ultrasonic bath for several hours before printing to ensure good homogeneity and to prevent sedimentation or aggregation of the particles. Then, the inks were filtered using a 5 μm filter to eliminate any large particles (Ag) or undissolved polymer (PEDOT:PSS) and hence avoid blocking the nozzles. Then, the cartridge was filled with ink and different numbers of layers were printed. In the case of the Ag ink, the printed patterns were cured at 120 °C for 5 min, and curing at 70 °C for 3 h was performed for the PEDOT:PSS ink. In this work, a ceramic (inorganic coating, 60 nm pore size) coated paper was used as a substrate and was purchased from PEL.

The quality and thickness of the inkjet printed silver and PEDOT:PSS stand-alone multi-layers were assessed using an optical microscope (LAOPHOT-2) fitted with a Nikon camera DS-Fi1 (Nikon, Tokyo, Japan) and scanning electron microscopy (SEM). Adhesive tape tests were performed in order to understand the adhesion properties of the Ag and PEDOT:PSS printed patterns to the selected substrates.

*Printing a full two terminal memory device*. The memory cells fabricated and investigated in this work were deposited on ceramic coated paper. The conductive tracks of Ag (5 passes) were first printed on the paper to define the bottom electrode (BE) of the final devices. After printing the Ag conductive paths, curing took place at 120 °C for 5 min. Then, the PEDOT:PSS active layer (10 passes) was deposited onto a paper marked with Ag tracks and then cured (70 °C, 3 h). Finally, in order to achieve a crossbar architecture, the top electrode (TE)–Ag (5 passes) was printed. The current–voltage (*I–V*) characteristics and retention time tests for the two terminal memory cells were measured in a screened sample chamber in the dark at room temperature using a PC-driven HP4140B picoammeter.

## 3. Results and Discussion

Wetting behaviour can be described as the interaction between the ink droplets and substrate. It should be highlighted that the printed quality is linked to the wettability of the ink and hence, the wettability of both inks on different substrates was investigated. In this study, four potential substrates were tested and the contact angle measurement results are presented in Figure 2. Two rigid (silicon and glass) and two flexible (paper and polyethylene terephthalate (PET)) substrates were studied. The paper substrate used in this work is ceramic coated (non-organic) and can be heated to 150 °C with minimal discolouration.

In addition to ensuring surface wetting, (i.e., ink/substrate perfect match), the surface energy of the substrate should exceed the surface tension of the ink by 10–15 mN/m. The surface energy of the ceramic coated paper was reported as 45 mN/m [24] and corroborates with the surface tension values (Table 1). Both inks were compatible with the ceramic coated paper. Thus, this substrate was selected to be used in this work. Furthermore, one of the main features of this substrate is that the ceramic layer helps to absorb the solvent from the ink. More specifically, in the case of Ag ink, the incorporation of the nanoparticles into the ceramic coating results in the formation of ceramic/Ag composites during the sintering process.

We have further investigated the spreading behaviour of both the silver and PEDOT:PSS inks (i.e., the evolution of the contact angle with time). As can be seen in Figure 3, in the case of the Ag ink the decrease of the contact angle takes place in two steps, one at 5 s and another at 10 s.

After 10 s of contact between the ink droplet and the substrate, the contact angle is quite constant with a value of 38° for the Ag ink (Figure 4). Interestingly, for the PEDOT:PSS ink the evolution of the contact angle is smoother, and there is no large variation between the first drop and the drops immediately following it. The behaviour observed for both of the inks is quite different. These differences arise due to the different constituents of the inks, as was discussed in the experimental section. Furthermore, the drop–surface interaction is a complex process that is governed by different forces, such as inertial, capillary and gravitational forces, a comprehensive study of which was published by Derby [25].

We suggest that the decrease in the contact angle over time may be associated with the rapid infiltration of solvent into the porous paper substrate controlled by capillary forces. This effect is more evident in the case of the PEDOT:PSS ink, which has only one solution phase and hence absorbs into the substrate more easily.

**Ag Printing Pattern.** In the case of a single printed Ag layer (one pass), a large number of disconnected particles can be seen in the optical microscopy image (Figure 5a). However, when more layers were printed, better connectivity or coverage is observed, as expected, as shown in Figure 5b. It is important to highlight that, along with the printability of the inks and the quality of the printed layers, another very important characteristic is the electrical behaviour of the printed patterns. Indeed, the continuity of the printed multi-layers has a crucial impact on the electrical properties (i.e., the conductivity of the printed-pattern). As expected, the patterns made by fewer passes were not electrically conductive, mainly due to the poor coverage or continuity of the silver particles, and hence were below the percolation threshold. However, in the case of five passes the coverage was much better and the resistivity was around 2.91 × 10^−8^ Ω·m, which is close to the values which have been reported previously [26].

The thickness of the printed layers was evaluated by taking a cross-sectional optical image and was found to be in the range of 1.1 and 1.8 µm for five passes (Figure 5d). Interestingly, the SEM image revealed penetration of the solvent into the paper [27], and hence validated our statement relating to the fast decrease of the contact angle.

**Printing PEDOT:PSS.** The correlation between the number of passes and the quality of the printed layers was evaluated by optical microscopy.

Both the optical microscopy and SEM images of the PEDOT:PSS ink revealed excellent coverage. Figure 6 clearly shows the high quality of the PEDOT:PSS printed pattern, and we postulate that the observed results are completely determined by the nature of the functional material (i.e., the polymer-based solution). Thus, the ink is homogenous without any dispersion component and the wetting is therefore better when compared to ink containing particle dispersions (Ag). In addition, the value of the surface tension of the PEDOT:PSS ink is slightly smaller than that of the Ag based ink and hence, better wetting behaviour is observed.

Furthermore, the electrical properties of the PEDOT:PSS ink patterns after 5 and 10 passes were investigated. The printed patterns display high resistive behaviour with resistance values of 1 GΩ for 5 passes and 50 MΩ for 10 passes. It is quite clear that an increase in the number of printing passes results in a decrease in the electrical resistance. This effect has also been observed by Sankir [28].

Additionally, the adhesion behaviour of the printed patterns on a paper substrate was tested. Good adhesion between the printed patterns and the substrate is an essential requirement for achieving optimal performance and reliability in this technology, and hence for real potential exploitation of inkjet printing technology in flexible electronics. The adhesion tape test was performed for both PEDOT:PSS and silver ink patterns. In this test, an adhesive tape is applied to the surface and then pulled off. The patterns were evaluated in the same area before and after the tape was removed. First, the adhesion features of the PEDOT:PSS ink pattern after one pass were evaluated. As can be seen in Figure 7a,b, no delamination of the printed pattern is observed. On the other hand, the pattern with five printed layers delaminates partially, as is illustrated in Figure 7c,d. Interestingly, the detached part is covered in paper, indicating that there is a strong cohesion among the PEDOT:PSS printed layers and weak adhesion between the paper and PEDOT:PSS, which could cause this failure. The strong cohesion among the printed PEDOT:PSS layers is very important for its applications to electronic devices, as it is acting as a single unit. Secondly, the experimental results show that by increasing the number of printed layers the cohesive force between the substrate and the printed pattern decreases.

Furthermore, no delamination was observed when the adhesion tape test was performed on the silver pattern (five passes), and images of the pattern before and after the test are presented in Figure 8a,b, respectively. Interestingly, as can be seen in Figure 8c, in this case the adhesive layer from the tape has been transferred and attached to the top of the printed Ag. In this situation, both the cohesion force between the silver layers and the adhesive force between the deposited layers and the solid substrate surface were very strong. Martínez-Sánchez et al. [29] demonstrated that the inclusion of silver nanoparticles in ceramic greatly enhances the mechanical properties, i.e., fracture toughness, while maintaining the levels of hardness and elasticity. We conclude that the high adhesion strength is due to the formation of ceramic–silver composites during the sintering process.

**Fully Printed Two Terminal Memory Devices.** Two terminal crossbar structure memory elements were fully fabricated at room temperature on a flexible substrate using only a commonplace desktop inkjet printer. Figure 9a shows an optical microscopy image of the fabricated crossbar array. First, in order to validate that our printed devices exhibit memory behaviour, the current–voltage (*I–V*) characteristics were analysed. Typical *I–V* behaviour can be observed for five consecutive scans of the Ag/PEDOT:PSS/Ag memory cell, as can be seen in Figure 9b. The printed device shows hysteresis in the *I–V* behaviour, which is an indication of electrical bistability in these devices. Additionally, *Vset*, *Vreset* and *Vread* have been identified from the *I–V* curves, and these values were further used when the memory retention time test was performed. The low conductivity state was programmed by applying one pulse at +10 V with a 1 ms width. The state was then read by decreasing the voltage to 6 V, where it was held for 4000 pulses while the current was monitored. Then, by applying a −8 V pulse for 1 ms the device was switched to a high conductivity state and the state was then read at 6 V. The high and low conductivity states remained distinguishable, as can be seen from Figure 9c.

In a nutshell, both the current–voltage and retention time tests validate that the fully printed devices are indeed exhibiting memory behaviour.

## 4. Conclusions

In this paper, we demonstrate that inkjet printing technology is capable of depositing full two terminal crossbar memory elements on a flexible substrate. Two functional materials, i.e., silver and PEDOT:PSS, as well as Ag/PEDOT:PSS/Ag structures were deposited by a commonplace Epson Stylus P50 desktop printer (Epson, Suwa, Nagano, Japan). The quality of the individual Ag and PEDOT:PSS printed patterns was investigated. It was found that in the case of the nanoparticulate ink the coverage was improved by increasing the number of printed layers. However, for the polymer, i.e., PEDOT:PSS, based ink a good surface profile was achieved even for a low number of passes. The cross-sectional image of the printed layers indicated that the structure was continuously built, with no sign of a barrier between individual passes being observed. The electrical properties of both of the printed patterns were investigated and the expected conductive behaviour of silver and high resistance for PEDOT:PSS were observed. Furthermore, adhesion tests for both inks were conducted. The experimental results showed excellent adhesion between the silver patterns and the flexible paper substrate. This is an effect of the ceramic–nanoparticle composites formed during the sintering process. On the other hand, the adhesion between PEDOT:PSS and the substrate is not desirable, and further work is required. Furthermore, a fully printed crossbar memory array was fabricated and the electrical behaviour of these cells was investigated.

We envision that this study will provide a novel platform for the deposition of functional materials and electronic devices, which has the potential to gradually replace conventional subtractive technologies and make a step forward to greener electronics.

## Figures and Tables

**Figure 1 micromachines-10-00417-f001:**
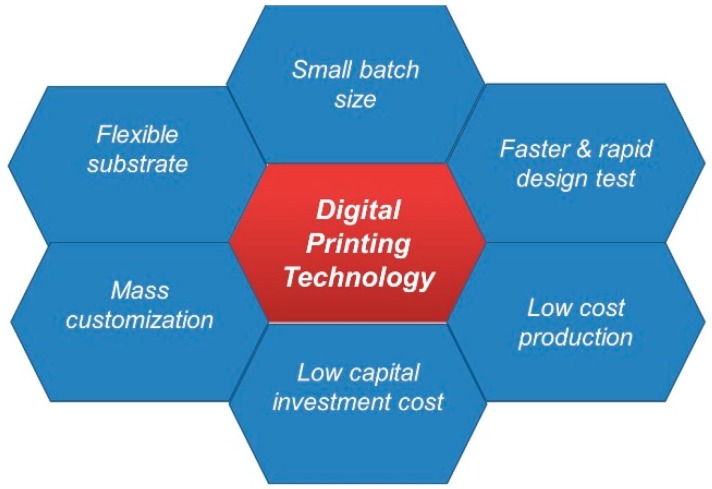
Main features of digital manufacturing technologies.

**Figure 2 micromachines-10-00417-f002:**
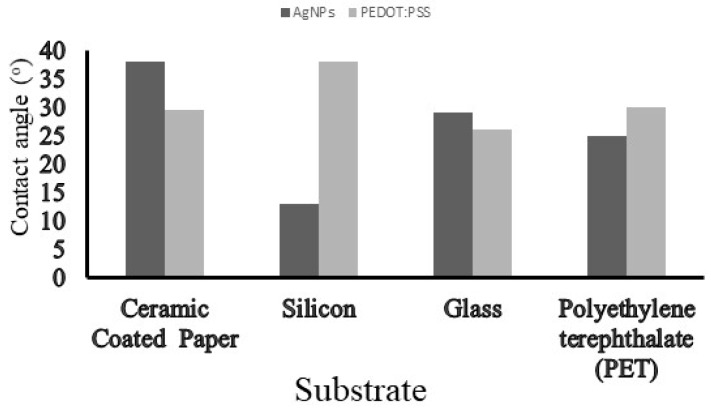
Contact angles measured via the sessile drop analysis method for the silver nanoparticles (AgNPs) and poly(3,4-ethylenedioxythiophene)-poly(styrenesulfonate) (PEDOT:PSS) inks and four potential substrates. The contact angle can be easily influenced by the nature of the substrate, mainly the surface.

**Figure 3 micromachines-10-00417-f003:**
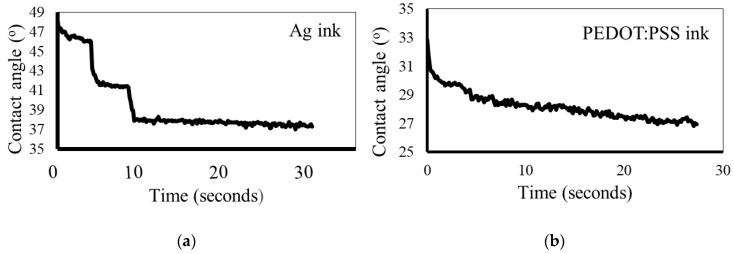
Contact angle versus time for the (**a**) Ag ink and (**b**) PEDOT:PSS ink on paper.

**Figure 4 micromachines-10-00417-f004:**
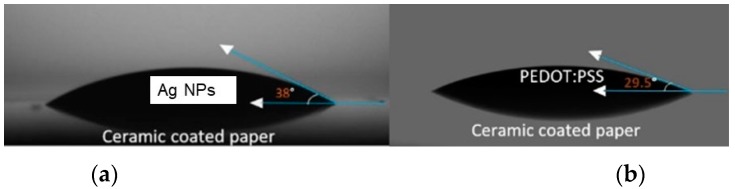
Images of the (**a**) AgNP and (**b**) PEDOT:PSS ink droplets on ceramic coated paper after achieving equilibrium during the contact angle measurements.

**Figure 5 micromachines-10-00417-f005:**
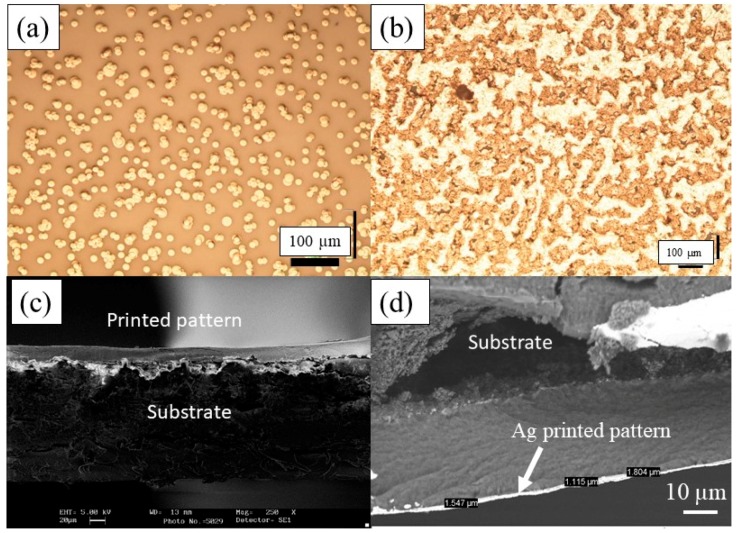
Optical micrograph of the inkjet printed Ag pattern after (**a**) 1 pass and (**b**) 10 passes. (**c**) SEM image of 10 passes. (**d**) SEM cross-sectional image of the Ag printed pattern after 5 passes.

**Figure 6 micromachines-10-00417-f006:**
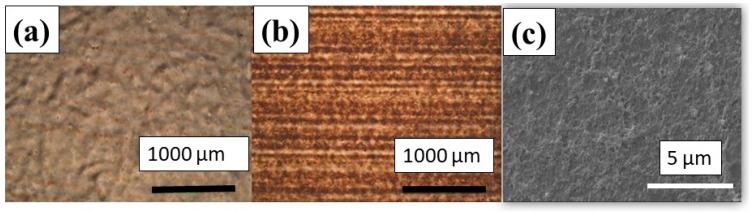
Optical micrograph of the inkjet printed PEDOT:PSS pattern after (**a**) two passes and (**b**) five passes. (**c**) SEM image of 5 passes of PEDOT:PSS ink pattern.

**Figure 7 micromachines-10-00417-f007:**
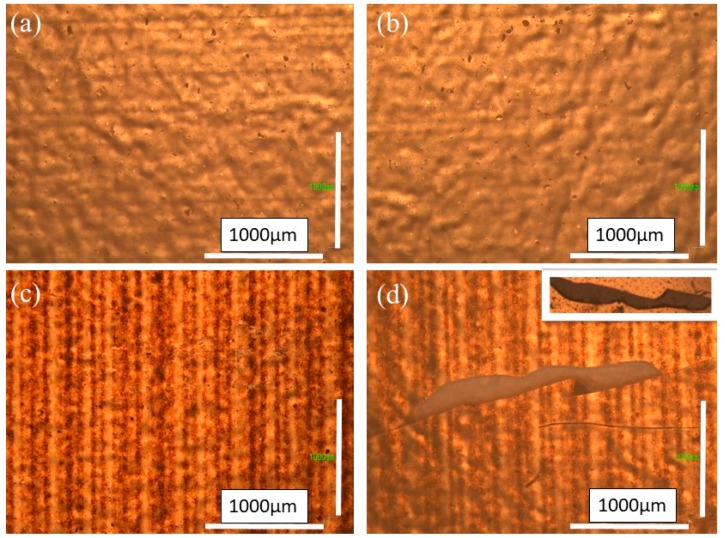
Optical microscopy images of the PEDOT:PSS ink pattern after one pass (**a**) before and (**b**) after removal of adhesion tape, and after five passes (**c**) before and (**d**) after removal of the adhesive tape.

**Figure 8 micromachines-10-00417-f008:**
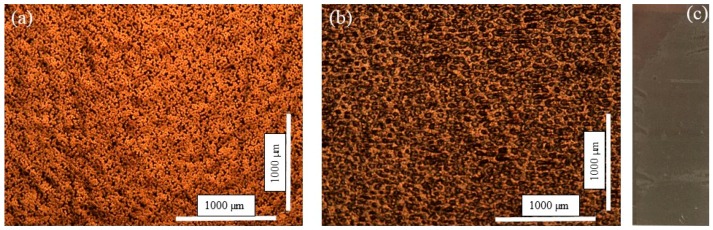
Optical microscopy images of printed Ag patterns after five passes (**a**) before and (**b**) after the removal of the adhesion tape. (**c**) Photograph of the Ag surface where the adhesive part of the tape was attached.

**Figure 9 micromachines-10-00417-f009:**
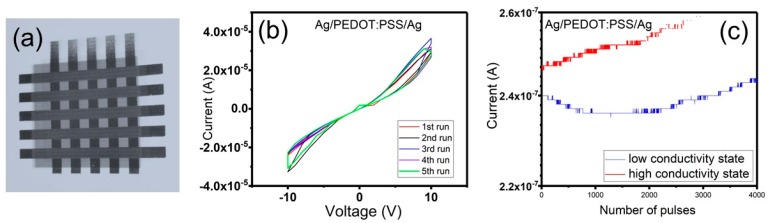
(**a**) Optical microscopy image of the printed crossbar memory cells. (**b**) *I–V* characteristics of five consecutive runs of the Ag/PEDOT:PSS/Ag memory elements. (**c**) The memory retention time of the printed cells.

**Table 1 micromachines-10-00417-t001:** Measured parameters of the Ag and PEDOT:PSS inks.

Parameters	Silver Ink	PEDOT:PSS Ink
Surface tension (mN/m)	35	30
Viscosity (cP)	6	14
pH	9	2.9

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
