# Peer review of "Inkjet Printing of Functional Electronic Memory Cells: A Step Forward to Green Electronics"

_micromachines, 2019, doi:10.3390/mi10060417_

Round 1

Reviewer 1 Report

This manuscript gives a detailed description of inkjet printing AgNP ink and PEDOT:PSS ink and then memory cells are also printed using low-cost inkjet printer. This manuscript looks interesting and can be accepted after minor revision.

1# In Abstract, authors mentioned "environmental issues". That should be fine. But I think the main purpose of developing printed electronics is due to the low-cost manufacturing with high throughput and relative ease of integration.

2# In 2. Experimental, authors use "silver (NPs Ag) ink". In fact, "AgNP ink"  is a more widely used abbreviation.

3# The optical images in both Fig.5d and 6d should be rotated 180 degrees to let the Ag pattern and PEDOT: PSS on top of the images.

4# Figure 5b is Ag pattern with 10 printing passes and Figure 8 is Ag pattern with 5 printing passes. However,  the pattern with 5 printing passes looks much dense. Why? due to the different magnification of the images?

5# The substrate used here is "PEL paper". Is PEL the brand or abbreviation of some kind of paper?

6# Printing technology or printed electronics is a thriving field. Printing on flexible (like paper substrate used in this manuscript) or stretchable substrate has drawn much attention. Some recent papers should be cited in Introduction part, such as Adv. Mater. Technol. 2019, 4, 1800546,  Small 2014, 10, 3515, Coatings 20188(8), 278; https://doi.org/10.3390/coatings8080278

Author Response

First of all, we would like to thank the reviewer for the providing suggestions/comments, as these allowed us to further enhance the quality of the manuscript. The following comments have now been fully addressed:

Reviewer 1:

This manuscript gives a detailed description of inkjet printing AgNP ink and PEDOT:PSS ink and then memory cells are also printed using low-cost inkjet printer. This manuscript looks interesting and can be accepted after minor revision.

We wish to thank the referee for the providing comments and the finding our work looks interesting.

1# In Abstract, authors mentioned "environmental issues". That should be fine. But I think the main purpose of developing printed electronics is due to the low-cost manufacturing with high throughput and relative ease of integration.

Indeed, one of the major aim of developing printing electronics is to reduce the manufacturing cost. However, the current technology to fabricate electronic devices requires heat generation in a deposition process and hence generation of harmful chemicals/radiation. Additionally, there are environmental limitations, for example, high vacuum equipment requires enormous amounts of electricity, thus creating a larger carbon footprint; alternatively, they use toxic gases in the production. The volume of waste material is also an environmental concern using conventional manufacturing technology. For example, using a spin coating method 80% of the material is a waste product. Based on all above mentioned facts we believe that inkjet printing offers a reliable alternative to traditional electronic manufacturing and most importantly, it is a solution to minimize the negative effects on the environment and to human health.

2# In Experimental, authors use "silver (NPs Ag) ink". In fact, "AgNP ink"  is a more widely used abbreviation.

Amended accordingly

3# The optical images in both Fig.5d and 6d should be rotated 180 degrees to let the Ag pattern and PEDOT: PSS on top of the images.

Due to the limitation of the optical microscope to collect/measure the correct thickness of the printed patterns those pictures have been removed from the revised manuscript and SEM image has in inserted.   Even more, the required changes are very difficult to operate as all the notations are made using the microscope’s software and if we like to rotate everything including numbers and text will be up-side down.

4# Figure 5b is Ag pattern with 10 printing passes and Figure 8 is Ag pattern with 5 printing passes. However,  the pattern with 5 printing passes looks much dense. Why? due to the different magnification of the images?

Yes, the different density of the printed patters is due to the different magnification used to collect the images.

5# The substrate used here is "PEL paper". Is PEL the brand or abbreviation of some kind of paper?

The “PEL paper” has been replaced by ceramic coated paper. Printed Electronics Limited (PEL) is the company from where this substrate was purchased.

We have provided/included in the Experimental part a short description of the paper substrate that has been used in this work as substrate.

Page 3_paragraph_4_line114/116: “In this work a ceramic (inorganic coating – 60 nm pore size) coated paper was used as substrate and was purchased from Printed Electronics Limited (PEL)”

6# Printing technology or printed electronics is a thriving field. Printing on flexible (like paper substrate used in this manuscript) or stretchable substrate has drawn much attention. Some recent papers should be cited in Introduction part, such as Adv. Mater. Technol. 2019, 4, 1800546,  Small 2014, 10, 3515, Coatings 20188(8), 278; https://doi.org/10.3390/coatings8080278

Thank you to reviewer for this suggestion. The recommended references have been added:

1.    Naghdi, S.; Rhee, K.; Hui,D.; Park, J.S; A review of conductive nanomaterials as conductive, transparent, and flexible coatings, thin films, and conductive fillers: Different deposition methods and applications, Coatings, 2018, 8(8), 278.

2.    Kamyshny, A.; and Magdassi, S.;  Conductive Nanomaterials for Printed Electronics, Small, 2014, 10, 3515.

3.    Huang, Q.; Zhu, Y.; Printing Conductive Nanomaterials for Flexible and Stretchable Electronics: A Review of Materials, Processes, and Applications, Adv. Mater. Technol. 2019, 4, 1800546.

Reviewer 2 Report

Thank you for your article and your investigations. I'd like to recommend some aspects to you:

PEDOT:PSS can be really used for memory applications. From this perspective I miss article SCIENTIFIC Reports | 6:19594 | DOI: 10.1038/srep19594 because here are explanations about what kind of memory effects can be used. Usually PEDOT:PSS is used as conductor but not any memory material. In this respect I miss the discussion about hysteresis of usual PEDOT:PSS because there are also effects know - sometimes simply caused by more or less water content.

The chose of an EPSON printer is possible but not the adequate choice. In your introduction you mention 'waveform'. This cannot be used-controlled in this printer. Therefore, a print setup like Dimatix, Ricoh, Konica Minolta, ... would be recommended. For low budget reserarch you should give at least any statement about this printer's waveform handling. Maybe you've adjusted the inks' properties to match the requirements of this printer? I cannot find any hint in this respect in your article.

When describing the printing there is just given 5 passes or 10 passes. There is no information about resolution, drop distance, drop diameter, printing speed and such kind of parameters. E.g. with Fujifilm Dimatix DMP 2831 a common layer thickness is in the range of 200 .. 400 nm for silver or PEDOT:PSS. I.e. 5 pass printing will add up to 2 µm. In your article you state about 10 times higher layer thicknesses. This requires explanation.

The buzzword "green electronics" in the headline is not reflected in the article. There a various kinds of meanings from non-chemical ... bioresorbable. And silver as material in going to be banned from mass-utilization - in this respect it isn't "green". Therefore, if you do not reflect this aspect in your article it shouldn't be included in the headline.

About State-of-the-Art: the Norwegian company Thinfilm already launched a printed memory device on the market. I would recommend that you at least mention these activities in your paper and compare your work to the commercial solution: what are different approaches and which technology offers which advantages/disadvantages.

Line related to your article

Sometimes a space looks like tow spaces: e.g. lines 10, 21, 77, 80, ... No idea for the reason

A space between value and unit is mandatory: lines123, 124, 136, 206, 253, 254, 255

17 - please double-check "and"

Figure 1 - "Disposable electronics" - this is not discussed in any way. Sometimes a vacuum processed part might have a better ecological footprint that a printed silver line?

61 - silver ist the best material: in which sense? Gold is better in functionality.

87 - ... via inkjet printing only in this work. I think you are wrong. There are dozends of articles describing inkjet printing of either silver and/or PEDOT:PSS. Maybe you try to give a somhow different statement?

124 - passses

127 - memeory

147 thissubstrate

150 i.e - "." missing

155 description is separated from figure

158 seconds

174 - start new sentence

183/184 description is separated from figure

185 please double check the thickness of layer. It's in contradiction to other literature sources. If you find supporting information, please insert references.

187 "reveals a penetration of solvent" How can this be proven? Solvents usually evaporate. Especially in the vacuum of an SEM.

192 Figure 6 c) not readable scale

203 please also double check 27 µm

231 transferred

241 Figure 8 a)b) Why is silver reddish? I would expect this from copper.

241 Figure 8 c) It's unclear what I see. Is it glue tape convering anything?

249/250 hysteresis in I-V is a feature of PEDOT:PSS, especially connected with water content

252/253 please keep value and unit on same line

359 functional

Author Response

First of all, we would like to thank the reviewer for the providing suggestions/comments, as these allowed us to further enhance the quality of the manuscript. The following comments have now been fully addressed:

PEDOT:PSS can be really used for memory applications. From this perspective I miss article SCIENTIFIC Reports | 6:19594 | DOI: 10.1038/srep19594 because here are explanations about what kind of memory effects can be used. Usually PEDOT:PSS is used as conductor but not any memory material. In this respect I miss the discussion about hysteresis of usual PEDOT:PSS because there are also effects know - sometimes simply caused by more or less water content.

We believe that the memory effect observed in our devices is based on electrical induce conformational changes of the active core. As PEDOT:PSS is an ionic compound, an electrical field can induce dipole-dipole or dipole charge interaction. However, in order to validate our statement further electrical and chemical investigations should be performed which is clearly beyond the scope of this paper.  

The chose of an EPSON printer is possible but not the adequate choice. In your introduction you mention 'waveform'. This cannot be used-controlled in this printer. Therefore, a print setup like Dimatix, Ricoh, Konica Minolta, ... would be recommended. For low budget reserarch you should give at least any statement about this printer's waveform handling. Maybe you've adjusted the inks' properties to match the requirements of this printer? I cannot find any hint in this respect in your article.

We thank to reviewer for raising this important point. The main aim of this work is to demonstrate that using a commonly used desktop printer is possible to fabricate a crossbar memory arrays. Indeed, we did not have access to setup of the waveform. However, using a £50 printed and by right selection/adjustment of the inks it’s possible to fabricate  memory cells. Further details  are provided in the revised manuscript.

Both the printer and silver ink were purchased from Printed Electronics Limited (PEL). Moreover, PEL optimized the ink to match the requirements of the printer.   

When describing the printing there is just given 5 passes or 10 passes. There is no information about resolution, drop distance, drop diameter, printing speed and such kind of parameters. E.g. with Fujifilm Dimatix DMP 2831 a common layer thickness is in the range of 200 .. 400 nm for silver or PEDOT:PSS. I.e. 5 pass printing will add up to 2 µm. In your article you state about 10 times higher layer thicknesses. This requires explanation.

In the first instance, we used optical microscope to measure the thickness. Unfortunately due to the limited resolution of the optical microscope the values of the measured thicknesses are not accurate and the figure 5d and 6d were removed from the revised manuscript. Furthermore, SEM was then used to measure the thickness of silver 5 passes and indeed the thickness is found to be in between 1.1 – 1.8 µm. The correct values are now included in the revised manuscript.

Additional information about the resolution, drop size are also  added in the manuscript.  

The buzzword "green electronics" in the headline is not reflected in the article. There a various kinds of meanings from non-chemical ... bioresorbable. And silver as material in going to be banned from mass-utilization - in this respect it isn't "green". Therefore, if you do not reflect this aspect in your article it shouldn't be included in the headline.

Indeed the reviewer is right if we discuss only at the materials used, however,  we used the “green electronics” terms to describe the advantages/impact of inkjet printing over other techniques like: thermal evaporation and spin coating that conventionally are used to fabricate this type of memory cells and not specific to the materials used. Thermal Evaporator requires enormous amounts of electricity, thus creating a larger carbon footprint; alternatively, they use toxic gases in the production and a large amount of time spend. The volume of waste material is also an environmental concern using conventional manufacturing technology. For example, using a spin coating method 80% of the material is a waste product.

About State-of-the-Art: the Norwegian company Thinfilm already launched a printed memory device on the market. I would recommend that you at least mention these activities in your paper and compare your work to the commercial solution: what are different approaches and which technology offers which advantages/disadvantages.

As the reviewer has suggested, we contacted the Thinfilm Company.  The memory product (the flexible, rewritable ferroelectric polymer-based memory) originally developed by Thinfilm, has been transferred to Xerox Corp for scale-up and manufacturing. However, based on the  information provided it is clear that ferroelectric polymer has been used.  However, we are unable to establish that if they memory cell is based on transistor concept or two terminal.

            We have provided further details in the revised manuscript to comply with the reviewer’s suggestion.  

Line related to your article

Sometimes a space looks like tow spaces: e.g. lines 10, 21, 77, 80, ... No idea for the reason

Amended accordingly

A space between value and unit is mandatory: lines123, 124, 136, 206, 253, 254, 255
Amended accordingly

17 - please double-check "and"

Figure 1 - "Disposable electronics" - this is not discussed in any way. Sometimes a vacuum processed part might have a better ecological footprint that a printed silver line?

As “disposable electronics” has not been discussed any more in the MS,  we removed this term from figure 1

61 - silver ist the best material: in which sense? Gold is better in functionality.

This aspect has been  now better articulated in the text:

Due to the aforementioned problems, the nanoparticles based silver ink [11-14] has attracted a special attention due to the stability, compatibility with the required ink’s additives, relatively higher resistant to oxidation, higher electrical conductivity and a very competitive price and  hence based on all these factors, we can thus safely conclude that currently, silver is the best metal for printing conductive patterns.

87 - ... via inkjet printing only in this work. I think you are wrong. There are dozends of articles describing inkjet printing of either silver and/or PEDOT:PSS. Maybe you try to give a somhow different statement?

Here you wanted to say that in this work only an inkjet printing technology was used to fabricate the Ag and PEDOT:PSS patterns. The appropriate clarification is provided in the manuscript

 The Silver (Ag) and poly(3,4-ethylenedioxythiophene):polystyrene sulfonate (PEDOT: PSS) inks were evaluated and deposited  via inkjet printing only.

124 – passses

Amended accordingly

127 – memeory

Amended accordingly

147 thissubstrate

Amended accordingly

158 seconds

Amended accordingly

174 - start new sentence

Amended accordingly

185 please double check the thickness of layer. It's in contradiction to other literature sources. If you find supporting information, please insert references.

The thickness of the layers has been checked and the correct values based on SEM image was added in the revise manusript.

187 "reveals a penetration of solvent" How can this be proven? Solvents usually evaporate. Especially in the vacuum of an SEM.

The ceramic coated paper used as substrate in this work was purchased  from  PEL to absorb solvents and dispersion agents and thereby concentrating the nanoparticles and enhancing low temperature sintering.

Reference is added

Hsiao, W.; Hoath, S.; Martin, G.; Hutchings, I.; Chilton, N.; Jones, S.; Imbibition dynamics of nano-particulate ink-jet drops on micro-porous media, Proc Nanotech 2011 Conference, Boston, US, June 2011.

192 Figure 6 c) not readable scale

Amended accordingly

The scale is included in the bottom bar of the SEM picture.

203 please also double check 27 µm

Amended accordingly

231 transferred

Amended accordingly

241 Figure 8 a)b) Why is silver reddish? I would expect this from copper.

The reddish colour is due to the settings of optical microscope.  I inserted a picture for your attention. The DMU logo/name is also printed by Ag ink.

241 Figure 8 c) It's unclear what I see. Is it glue tape convering anything?

The fig. 3c shows the sticky layer was detached from the glue tape and has been attached to the Ag pattern. 

249/250 hysteresis in I-V is a feature of PEDOT:PSS, especially connected with water content

The PEDOT:PSS patterns are annealed after printing and we consider that the remaining water  completely evaporates during this  process. 

252/253 please keep value and unit on same line

Amended accordingly

359 functional

Amended accordingly
